# Are we missing the forest for the trees? Conspecific negative density dependence in a temperate deciduous forest

Kathryn E. Barry 👩🔬[1]*, Stefan A. Schnitzer[2]

**1** Department of Biology, Ecology and Biodiversity Working Group, Institute of Environmental Biology, Utrecht University, Utrecht, Netherlands, **2** Department of Biological Sciences, Marquette University, Milwaukee, WI, United States of America

* barry.kt@gmail.com

**Data Availability Statement:** All data and code for this project are now available in my public GitHub repository here: https://github.com/katie-barry44/barry-schnitzer2021PlosOne.

## Abstract

One of the central goals of ecology is to determine the mechanisms that enable coexistence among species. Evidence is accruing that conspecific negative density dependence (CNDD), the process by which plant seedlings are unable to survive in the area surrounding adults of their same species, is a major contributor to tree species coexistence. However, for CNDD to maintain community-level diversity, three conditions must be met. First, CNDD must maintain diversity for the majority of the woody plant community (rather than merely specific groups). Second, the pattern of repelled recruitment must increase in with plant size. Third, CNDD should extend to the majority of plant life history strategies. These three conditions are rarely tested simultaneously. In this study, we simultaneously test all three conditions in a woody plant community in a North American temperate forest. We examined whether understory and canopy woody species across height categories and dispersal syndromes were overdispersed–a spatial pattern indicative of CNDD–using spatial point pattern analysis across life history stages and strategies. We found that there was a strong signal of overdispersal at the community level. Across the whole community, larger individuals were more overdispersed than smaller individuals. The overdispersion of large individuals, however, was driven by canopy trees. By contrast, understory woody species were not overdispersed as adults. This finding indicates that the focus on trees for the vast majority of CNDD studies may have biased the perception of the prevalence of CNDD as a dominant mechanism that maintains community-level diversity when, according to our data, CNDD may be restricted largely to trees.

## Introduction

Conspecific negative density dependence (CNDD) is one of the most empirically supported mechanisms for the maintenance of plant species diversity [1–4]. Conspecific negative density dependence occurs when small individuals have relatively low rates of growth and survival near adult members of their own species (conspecifics). This constraint on growth near

**Funding:** This work was funded by two Rea Fellowships awarded by the Carnegie Museum of Natural History to K.E.B. Funding for field assistance was provided by a Research Growth Initiative from the University of Wisconsin-Milwaukee to S.A.S. Additional funding for K.E.B. was provided by the Ivy Balsam-Milwaukee Audobon Society Grant.

**Competing interests:** The authors have declared that no competing interests exist.

conspecifics results in a distinct spatial pattern where adult conspecifics occur further away from each other than would be expected by chance (overdispersion; [5]; but see [6,7]). Thus, CNDD is predicted to result in stable species coexistence across the landscape because dominant species cannot displace subordinate ones [8,9]. Evidence for CNDD has been reported in a variety of ecosystems, including lakes, deserts, grasslands, marine ecosystems, and particularly in temperate and tropical forests [1,2,7,10–16]. For forests, over the past ten years alone, evidence for CNDD has been reported more than 30 times in 13 countries across five continents [17].

However, strong evidence for CNDD at the seedling level may not result in maintenance of diversity at the forest level if NDD reduces clustering of seedlings but does not overcome the initial clumped pattern of seedlings around adults (due to dispersal limitation) [18]. That is, if CNDD does not result in a pattern of overdispersion, then it may fail to stably maintain species diversity ([5,19,20]; but see [21,22]). For CNDD to be a likely mechanism maintaining community-level diversity in temperate forests, the following three conditions must be met. 1) Individuals of the majority of the species in the community will be overdispersed because of greater mortality near conspecific adults. If only a small proportion of the species are overdispersed, then CNDD may not theoretically benefit rare species enough to maintain diversity [8,9]. 2) The degree of overdispersion will increase with ontogenetic (life-history) stage. That is, the signal of CNDD should compound as individuals mature, and thus larger individuals of any given species should be more overdispersed than smaller individuals (see Zhu *et al.* 2015 [18]). 3) CNDD will operate across life history strategies, including species that vary in growth form and dispersal syndrome. If CNDD is the main mechanism driving diversity maintenance, as suggested by previous studies (e.g., [3,17]) then it should operate on the plant growth forms that contain the highest diversity. If these three conditions are met, then CNDD is likely to be sufficiently strong to maintain community-level diversity.

Previous studies may have overestimated the importance of CNDD in forest ecosystems for two reasons. First, the vast majority of studies that examined CNDD in vascular plant species focused on growth and mortality at the seedling stage [1,2,14,16,23,24]. Dynamics at the seed-to-seedling and seedling-to-sapling transitions do not necessarily translate to overdispersion in the larger size classes [25] and may overestimate the role of CNDD [26]. Due to dispersal limitation, most seeds arrive beneath the parent tree and thus most seedlings are also congregated there. CNDD can maintain diversity only if seedling mortality beneath the conspecific is sufficiently strong to overcome and reverse the initial clumped seedling distribution. Furthermore, CNDD must be high enough to exceed the null expectation of high seedling mortality near the parent tree purely because there are more seedlings present [27]. If so, the negative effects of growing near a conspecific adult should compound as individuals mature. As individuals grow, they compete more intensely with adults or acquire more pathogens or both, and thus the level of overdispersion should increase with plant size.

Currently, the evidence for CNDD beyond the seed to seedling transition is mixed. For example, a study by Yao et al. [25] found that CNDD decreased with increasing tree ontogeny in a temperate forest. In fact, many species in both temperate and tropical forests do not have an overdispersed distribution [19,28]. By contrast, Guo et al. [29] found that 75% of tree species demonstrated CNDD as adults in subtropical forests (see also [30–32]).

Second, the vast majority of CNDD studies in forests focused only on trees, ignoring other important plant growth forms (e.g., [5]). The selection of trees to test CNDD as a general mechanism for the maintenance of diversity is particularly problematic in temperate forests, where canopy tree species represent a relatively small fraction (~7%) of the total vascular plant community [33]. Furthermore, canopy trees may be more prone to overdispersion due to their capacity for long distance dispersal [34–36] and thus they may bias our understanding of the

importance of CNDD for the diversity maintenance of the larger plant community. By contrast, understory plants (including understory woody species) represent a larger share of diversity but have a lower capacity for long distance dispersal due to their relatively short stature and position in forest understory. Furthermore, few understory plant species have dispersal syndromes that favor long distance dispersal [33]. Many understory species are gravity dispersed while the majority of temperate canopy trees are wind dispersed. Thus, the strength of CNDD may interact with plant dispersal syndrome.

Nonetheless, if CNDD is the primary mechanism that maintains community level diversity, we would expect it to operate across life history stage and life history strategy. We addressed these three core conditions for CNDD to be a general mechanism for the maintenance of plant species diversity by evaluating the spatial patterns of a woody plant community across life history strategies (shrubs, understory trees, mid-story trees, canopy trees, and lianas) and ontogenetic stages (seedling, sapling, and adult) in the field in a temperate forest in western Pennsylvania, USA. We tested three specific hypotheses: 1) Woody plant diversity in temperate forests is maintained by CNDD, and thus we predict that the majority of plant species will be overdispersed. 2) The effects of CNDD compound as plants grow, and thus overdispersion will increase with plant size. 3) CNDD operates independently of growth form and dispersal syndrome, and thus we predict that the pattern of overdispersion will be found in the majority of the species of all plant groups. We tested these hypotheses by examining the degree of overdispersion in a woody plant community, which included a range of plant life-history stages (*i.e.*, sizes) and life-history strategies (*i.e.*, growth form and dispersal syndrome).

## Materials and methods

### Study site

We conducted this study at Powdermill Nature Reserve with permission from the Carnegie Museum of Natural History. Powdermill Nature Reserve is an 890-hectare reserve located in the Allegheny plateau at the base of the Appalachian Mountains in southwestern Pennsylvania, USA (Westmoreland County; 40°09' N, 79°16' W). This region receives ~1100 mm of precipitation per year and is characterized by mixed mesophytic vegetation that is dominated by maples (*Acer spp.*), tuliptree (*Liriodendron tulipifera*), and oaks (*Quercus spp.*). Elevation at Powdermill Nature Reserve ranges from 392 to 647 m above sea level. Powdermill Nature Reserve contains a matrix of vegetation types consisting primarily of secondary deciduous forest but with several areas of maintained fields and managed lands. Last known logging occurred in this region in the 19th century, and land was primarily used for agriculture into the early 20th century (see [37] for more detailed site description).

### Plot establishment and plant census

In May and June of 2014, we established sixteen 10-m diameter circular plots in the >90-year-old secondary temperate deciduous forests at Powdermill Nature Reserve. We chose the 10-m diameter spatial grain because this size was thought to be a suitable size to test for spatial patterns associated with NDD in a Malaysian forest ([38]; see also [19,28]). We avoided canopy gaps for the placement of each plot, and each plot had >80% canopy cover. We ensured that the plots were not within 10 m of a waterway, that soil cover was not predominantly rocks, and that plots were at least 20 m from any edge. We used a Trimble GeoExplorer 6000XH to measure the precise location (up to 10 cm accuracy) of all woody plant individuals >10 cm height in each plot (Trimble Navigation Limited, Westminster, CO). For each individual, we measured height and basal diameter, and we identified them to species.

To examine how overdispersion changes with plant size, we divided individuals into four height classes (<0.5 m, 0.5–1 m, 1–5 m, and 5–10 m). We use these height classes as a proxy for both relative age (assuming that plants get taller as they get older) and position within the forest. Individuals that are shorter are less likely to be able to disperse seeds farther away than individuals that are taller even if they belong to the same species and have the same dispersal syndrome. To understand how overdispersion interacts with life-history strategy, we classified each species as either canopy or understory (growth form), and as either bird, wind, self, or other animal dispersed (dispersal mechanism) based on species descriptions in the Flora of North America [39].

## Data analysis

We performed all data analysis in R statistical computing software [40]. To measure plant spatial distribution (the degree to which plants are clustered or overdispersed), we calculated Ripley's K in the package "spatstat" using Ripley's translational border correction at each plot for each species and then, for ease of interpretation, converted K to Besag's L [41–44]. Several studies have demonstrated that spatial point pattern analysis is capable of detecting spatial patterns that can be attributed to mechanistic processes (e.g. [5,45]). To eliminate point patterns based on low replication, we removed species at any plot with fewer than five individuals as point patterns with fewer than 5 points in our data had significantly larger variance than those with >5 points. We then calculated a pooled L for each comparison (by species, by growth form, by growth form/plant size, or by growth form/dispersal mechanism) by weighting the individual L estimates by the number of points in a given L-function (methods follow [46]). For the growth form by dispersal mechanism interaction, we limited the final analysis to bird and wind dispersed species because these two groups had sufficient replication for robust comparisons between canopy and understory plants.

We bootstrapped these estimates 999 times to create 95% confidence intervals. We then calculated the predicted L for complete spatial random to compare our spatial patterns to complete spatial random. Data manipulation of input to and output from point pattern analysis was done using a combination of the "abind", "gridExtra", and "reshape" packages [47–49]. We constructed all figures in the package "ggplot2" [50].

To more easily interpret the figures, we corrected our measures of L with the distance at which each measure of L was calculated (L(d)-d). Besag's L is a measure of spatial aggregation, and when L(d)-d is positive, a greater proportion of neighbors are observed within distance d of focal individuals than predicted by a complete spatial random pattern. When L(d)-d is negative, a smaller proportion of neighbors are observed within distance d of focal individuals than predicted by a complete spatial random pattern (Fig 1, [51]). We considered plants to be overdispersed based on their point pattern when the linear regression slope of L(d)-d was significant and positive (using the command "lm" in R, S1 and S2 Tables) with increasing distance (d) [46]. This designation implies that more individuals are found far away from an individual of a given species than near an individual of that species. We considered plants to be clustered based on their point pattern when the linear regression slope of L(d)-d was significant and negative with increasing distance. These designations differ from "pure overdispersion" (i.e. regularity or inhibition), which would begin with a significantly negative L(d)-d that indicates fewer individuals close to the parent than would be expected by chance (Fig 1B and 1C, [44,46,52]). However, natural dispersal typically results in more conspecific seeds and seedlings close to adults than predicted by complete spatial random, and thus we did not expect to find a significant negative L(d)-d of seedlings close to the parent [53]. Therefore, we accounted for dispersal limitation by focusing on overdispersion as having a positive slope with regards

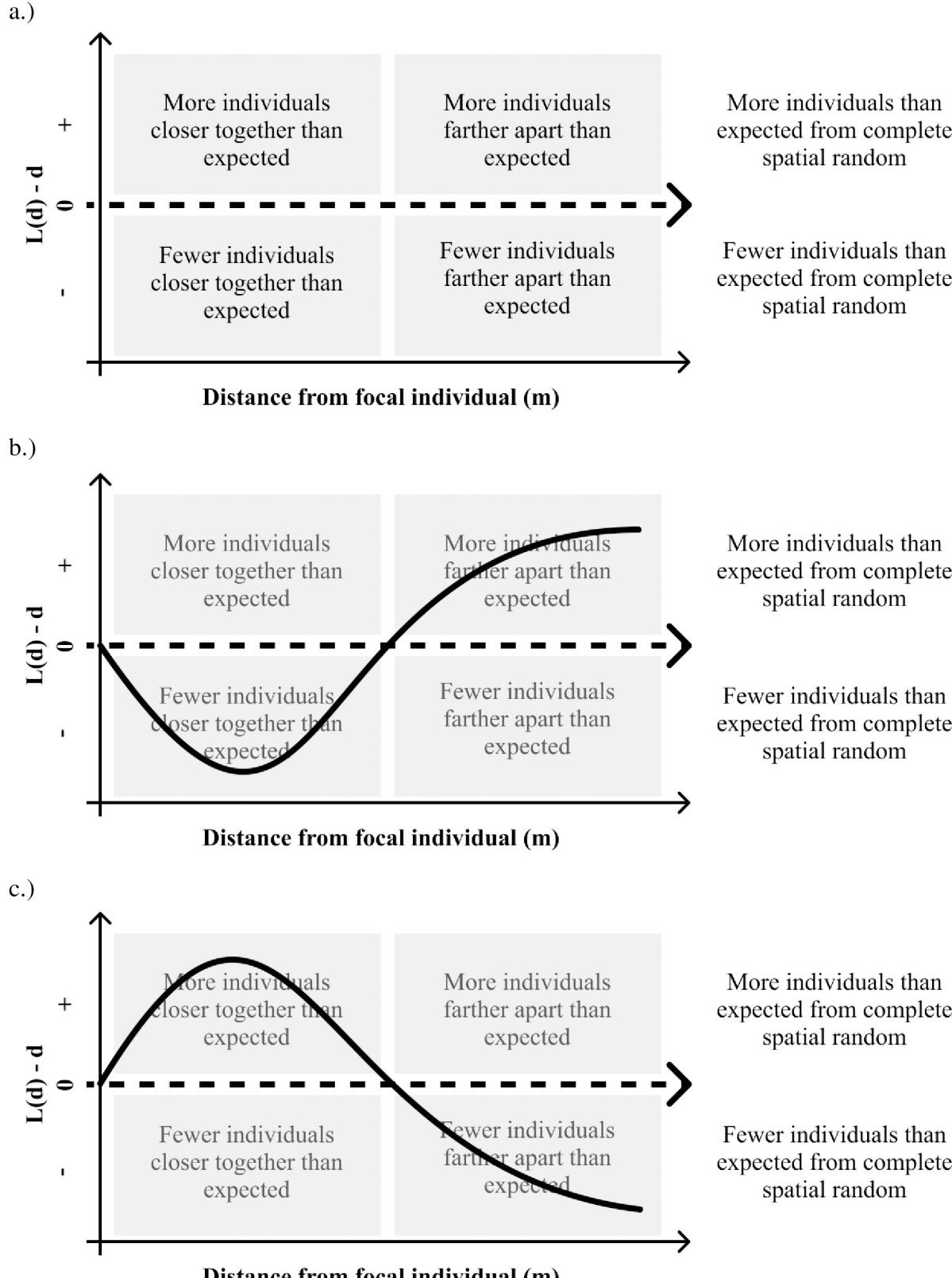

**Fig 1. Conceptual representation of L(d)-d and representative spatial patterns.** a.) Conceptual diagram of interpretations of different quadrants of spatial point pattern space for Besag's L after it has been centered by distance. b.) Hypothetical distribution where individuals

are perfectly overdispersed, there are more individuals than expected from complete spatial random but only at larger distances, i.e. those individuals are farther apart. c.) Hypothetical distribution where individuals are perfectly underdispersed. There are more individuals than expected from complete spatial random but only at shorter distances, i.e. those individuals are close together.

to distance (d), which indicates a significant increase in individuals with distance from the adult, at the community scale. However, CNDD should result in increasing overdispersion with plant size as the effects of CNDD compound with time (as the plant matures). Thus, we might expect a spatial signature of "pure overdispersion" in larger size classes if CNDD is capable of overcoming initial dispersal patterns and thus stably maintaining coexistence.

We considered any point pattern to be significantly different from complete spatial random if a mixed effect linear model including all factors of the L(d)-d and the distance was significant (calculated using the command "lmer" in package "lme4" with plot as a random effect and using the package "lmerTest" to calculate p-values; Table 1; [54,55]). Further, we include the results of the same models but with a more conservative estimate of degrees of freedom in S2 Table. If the total model was considered significant, we did not consider the point pattern to be significantly different from complete spatial random at any distance where the bootstrapped 95% confidence intervals of L(d)-d overlap with complete spatial random. We considered any two point patterns to be significantly different from each other if their bootstrapped 95% confidence intervals did not overlap at a given distance.

**Table 1. Results of mixed effects linear model to calculate L statistic significance at Powdermill Nature Reserve in Southwestern Pennsylvania for all comparisons of all individuals >10 cm height.**

| Model | dF | T Stat | P value | Figure |
|---|---|---|---|---|
| All individuals | 2228 | 19.37 | <0.0001 | 1a |
| All individuals, <0.5 | 781 | -4.477 | 0.6689 | 1b |
| All individuals, 0.5–1 | 889 | 7.83 | <0.0001 | 1b |
| All individuals, 1–5 | 797 | 8.5828 | <0.0001 | 1b |
| All individuals, >5 | 358 | -14.97 | <0.0001 | 1b |
| Understory | 1071 | 10.191 | <0.0001 | 2a |
| Canopy | 1059 | 5.796 | <0.0001 | 2a |
| Canopy, <0.5 m tall | 380 | -7.14 | <0.0001 | 2b |
| Canopy, 0.5–1 m tall | 478 | 8.67 | <0.0001 | 2b |
| Canopy, 1–5 m tall | 463 | 9.14 | <0.0001 | 2b |
| Canopy, >5 m tall | 186 | -10.71 | <0.0001 | 2b |
| Understory, <0.5 m tall | 397 | 2.045 | 0.0415 | 2c |
| Understory, 0.5–1 m tall | 407 | 0.3348 | 0.7379 | 2c |
| Understory, 1–5 m tall | 381 | 1.038 | 0.2998 | 2c |
| All individuals, bird dispersed | 1159 | 2.86 | 0.0042 | 3a |
| All individuals, animal dispersed | 200 | -6.806 | <0.0001 | 3a |
| All individuals, wind dispersed | 803 | 3.404 | <0.0001 | 3a |
| Overstory, wind dispersed | 601 | 3.903 | <0.0001 | 3b |
| Overstory, bird dispersed | 250 | 5.482 | <0.0001 | 3b |
| Understory, wind dispersed | 199 | -8.895 | <0.0001 | 3b |
| Understory, bird dispersed | 906 | -3.533 | 0.0004 | 3b |

To calculate significant differences from complete spatial random we used a mixed effects linear model with plot as a random effect to control for between plot differences due to environmental heterogeneity between plots. We report model degrees of freedom based on the number of L estimates (calculated every 10 cm per point pattern per plot). We report any pooled point pattern as overdispersed if it has a significantly positive slope and any pooled point pattern as clustered if it has a significantly negative slope.

## Results

At the community level, all woody plants combined were significantly overdispersed (Fig 2A). The largest individuals (>5 m tall) had a significantly lower overdispersion (L(d)-d) at intermediate distances (2-5m), than the two middle height size classes (1m – 5m and 0.5–1 m); however, L(d)–d did not differ significantly among the larger size classes at distances greater than 5 m (Fig 2B). By contrast, the smallest individuals (< 0.5 m) had significantly lower overdispersion than intermediate height individuals (0.5-1m and 1–5 m tall) for all distances greater than 2m (Fig 2B), and significantly lower dispersion than individuals in all of the larger height categories for distances greater than 5m. Thus, all but the smallest size classes were overdispersed at longer distances from the adult tree, indicating that, at the community-level, NDD was strong enough to overcome the initial clumped distribution of seedlings as the plants grew.

Both canopy trees and understory plants were significantly overdispersed; canopy trees were more overdispersed (significantly higher L(d)-d) at distances greater than 3 m (Fig 2A). The differences in overdispersion between canopy and understory plants become more pronounced with plant life history stage (*i.e.*, plant size). Canopy trees did not differ significantly from complete spatial random when they were small and young, but became significantly overdispersed when they were larger (Fig 3B), which is consistent with CNDD. Understory plants displayed the opposite pattern: they were overdispersed when small, but larger individuals were indistinguishable from complete spatial random (Fig 3C).

All of the four dispersal mechanisms that we examined, wind, bird, and self-dispersed species were overdispersed and statistically indistinguishable from each other. Species dispersed by animals other than birds (including secondary dispersal by squirrels) were all significantly less overdispersed than the other three dispersal types (Fig 4A). Dispersal syndrome for bird and wind dispersed species did not explain the differences in spatial pattern between canopy trees and understory plants; canopy trees were always more overdispersed than understory plants regardless of dispersal mechanism (Fig 4B), suggesting that the height of canopy trees is the most important factor in dispersal distance.

## Discussion

We found that canopy trees were overdispersed and the strength of overdispersion increased with tree size–two critical conditions for CNDD to be a general mechanism for the maintenance of woody plant species diversity. Increasing overdispersion with increasing plant size is predicted by CNDD because plants should survive and grow best away from conspecific adults due to intraspecific competition [56] or the negative effects of natural enemies [8,9,57,58]. Our findings are consistent with a growing number of studies that have reported that CNDD is a viable mechanism to maintain canopy tree diversity in temperate and tropical forests (*e.g.*, [1,2,7,14,32,38]). Thus, our findings support CNDD as a mechanism for the maintenance of canopy tree species diversity. However, we cannot rule out the possibility that trees may be more likely to be overdispersed with size simply because the larger (and presumably older) the tree the greater the probability of mortality for the parent (which is often nearby) [59].

For woody understory plants, our spatial patterns did not meet the criteria for CNDD to maintain species diversity. Understory species were overdispersed only in the smallest size classes, and overdispersion did not increase with plant size, which we use as a proxy for life history stage. If CNDD is operating in understory plants in these forests, it does not appear to be sufficiently strong to overcome the initial clumped dispersal pattern of seedlings, and therefore it did not result in overdispersion. Similar conclusions that CNDD may not be a general mechanism for the maintenance of non-tree plant diversity were reported for tropical forests. For

**a.)**

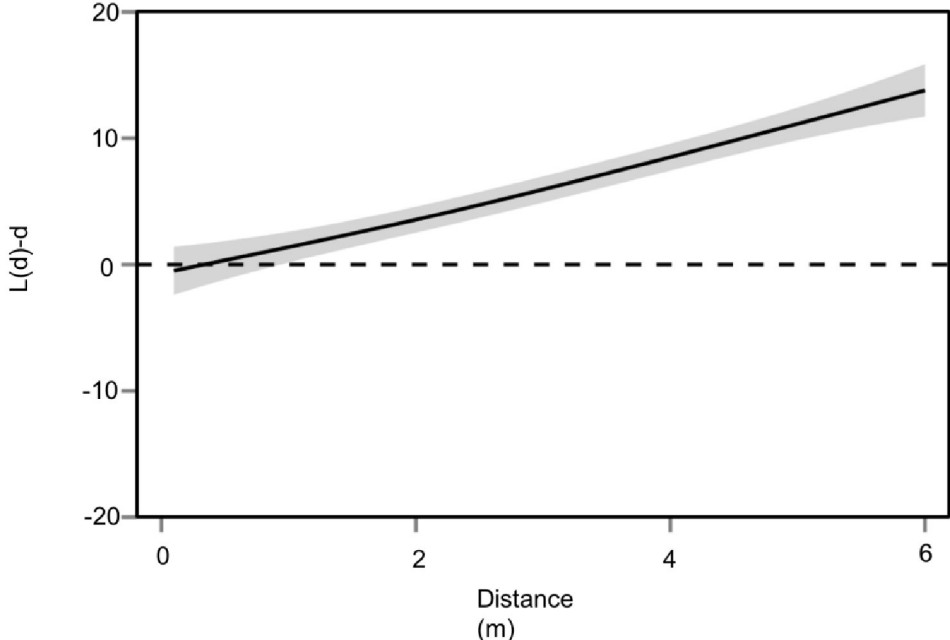

**b.)**

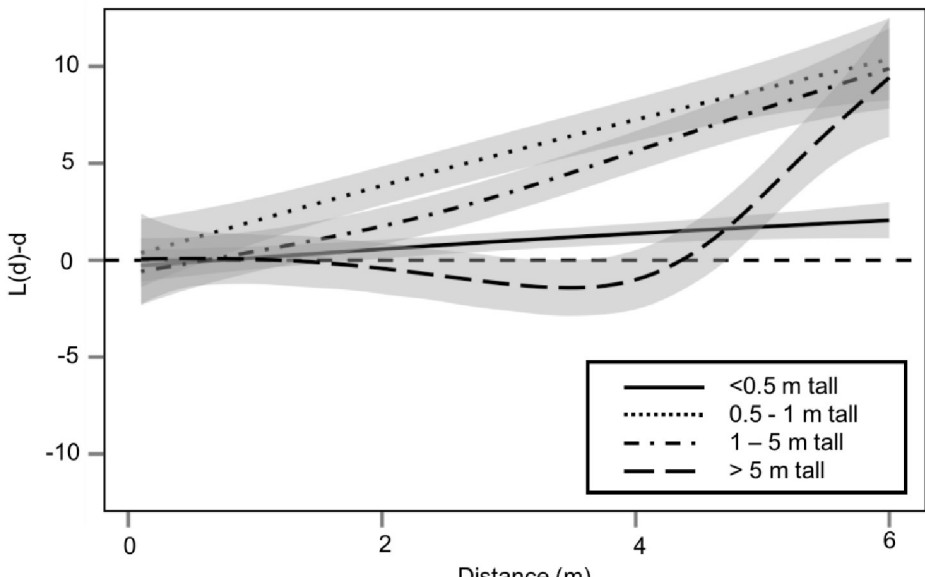

**Fig 2. Pooled Besag's L statistic across distance from spatial point pattern analysis for the full community of woody plants >10 cm in height at Powdermill Nature Reserve in Southwestern Pennsylvania.** a.) The community of woody plants (all species, n = 62 point patterns) was significantly overdispersed regardless of dispersal mechanism. However, the L(d)-d for the community remains positive across all distances indicating that some individuals occur close to members of their own species. b.) Individuals that were <0.5 m tall were the least overdispersed (n = 25 point patterns). Individuals that were intermediate in height (0.5m to 5 m tall) were significantly more overdispersed than smaller individuals, though not significantly more or less overdispersed than the largest individuals ($n_{0.5-1m}$ = 26 point patterns, $n_{1-5m}$ = 27 point patterns). The largest individuals (> 5m tall, n = 13 point patterns) were not significantly more overdispersed than individuals that were 0.5m to 5m tall; however, the drop in the line below complete spatial

random indicates that they had less clumping over small distances. Grey shaded regions represent 95% confidence intervals, darker grey regions represent overlapping confidence intervals. Overlap in 95% confidence intervals indicates that spatial point patterns were either not significantly different from each other (when two spatial point patterns overlap) or that a spatial point pattern did not differ from complete spatial random (when overlapping with the black dotted line).

example, Ledo & Schnitzer [5] found that lianas, which comprised ~35% of the woody species diversity in a Panamanian tropical forest [60,61], were underdispersed (clustered) rather than overdispersed. Thus, Ledo & Schnitzer [5] concluded that, while there was evidence for CNDD for canopy trees, there was little evidence for CNDD for lianas. Similarly, in a Caribbean tropical forest, DeWalt and colleagues [36] found that non-canopy tree woody seedlings (lianas and shrubs) were less likely to suffer negative density dependent mortality than canopy trees. In tropical forests, however, trees commonly represent 65% or more of the woody plant species diversity (*e.g.*, [60,61]), and thus CNDD is still likely a powerful diversity maintenance mechanism. By contrast, CNDD may fail to maintain the majority of species diversity in temperate forests where canopy trees represent a small minority of species [62,63].

In temperate forests, CNDD likely does not occur in isolation. Rather, CNDD and other mechanisms like facilitation, niche specialization, and dispersal limitation likely interact to maintain diversity in these forests. CNDD may be the most important mechanism for the maintenance of tree species diversity even though these other mechanisms are likely to be occurring simultaneously. But for other plant groups, these other mechanisms like facilitation, niche specialization, and dispersal limitation may be more important relative to CNDD. For example, Ledo and Schnitzer [5], found that clumped spatial distributions may be due to niche specialization in lianas, while trees demonstrated overdispersion indicating that CNDD may be more powerful. Similarly, the relative importance of these different mechanisms may change as plants grow. For example, Yao et al. [25] found that CNDD was important for individuals when they were young and small but that topographic and edaphic factors increased in importance with increasing plant age. Similarly, for tree seedlings invading into a grassland, Wright et al. [64] found that smaller tree seedlings benefited from facilitation in high diversity contexts while larger tree seedlings experienced strong competition.

At Powdermill Nature Reserve, a similar scenario where overall diversity is maintained by several mechanisms which simultaneously support diversity but also tradeoff in importance depending on the age/size of individuals and their abiotic context. Trees (and especially the largest trees) may be maintained largely by CNDD; whereas, understory plants may be influenced by a number of different mechanisms. There is evidence that CNDD is a weak mechanism for the maintenance of understory plant diversity, since overdispersion is present when understory plants are small (Fig 2C). However, the lack of overdispersion in larger understory plants indicates that a mechanism (or mechanisms) other than CNDD is a stronger driver of understory plant diversity. Short distance dispersal is often adaptive because site conditions are likely to be the same in the area immediately surrounding a parent plant [65]. Because dispersal syndromes that favor shorter distance dispersal are more common in the understory, mechanisms like niche differentiation that rely on adaptation to specific abiotic factors as found by both Ledo and Schnitzer [5] and Yao et al. [25] may be more important for these understory species.

Canopy trees may be significantly more overdispersed than understory species simply because being tall enables longer distance dispersal. We found higher overdispersion of canopy trees than for understory plants regardless of dispersal mechanism (Fig 4B). That is, tall species were more likely to experience overdispersion whether they were bird or wind dispersed even though bird dispersal may enable more (generally rare) events of very long-distance dispersal

a.)

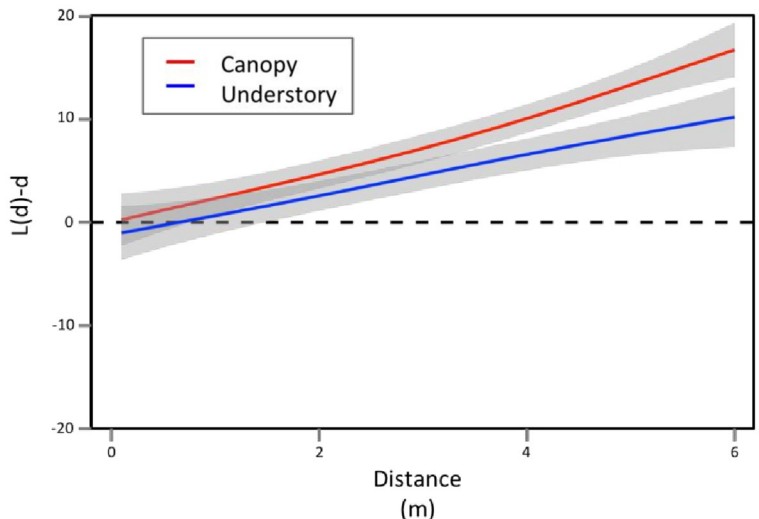

b.) Canopy species

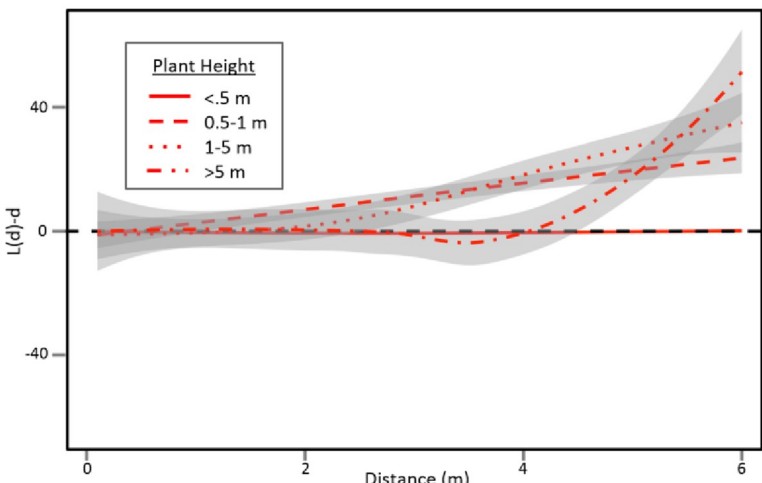

c.) Understory species

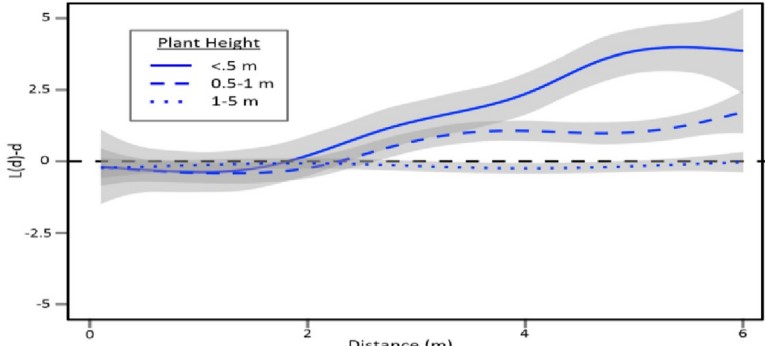

**Fig 3. Pooled Besag's L statistic across distance from spatial point pattern analysis for woody plants >10 cm in height at Powdermill Nature Reserve in Southwestern Pennsylvania separated by growth form.** Black dotted line throughout represents the complete spatial random prediction. a.) Canopy (n = 29 point patterns) and understory plants (n = 33 point patterns) were both significantly overdispersed, indicative of negative density dependence. Canopy plants were significantly more overdispersed than understory plants. b.) Canopy plants were more overdispersed with increasing life-history stage in accordance with predictions for negative density dependence ($n_{<0.5}$ = 14 point patterns, $n_{0.5–1}$ = 48 point patterns, $n_{1-5}$ = 32 point patterns, $n_{>5}$ = 15 point patterns). c.) Understory plants were not more overdispersed with life-history stage ($n_{<0.5}$ = 21 point patterns, $n_{0.5–1}$ = 22 point patterns, $n_{1-5}$ = 20 point patterns). Grey shaded regions represent 95% confidence intervals, darker grey regions represent overlapping confidence intervals. When confidence intervals overlap, we consider two point patterns to be the same in the overlapping region. We consider point patterns where the confidence intervals overlap with the black dotted line to not be significantly different from complete spatial random in that region.

[66,67]. Understory plants tend to have universally smaller dispersal kernels regardless of dispersal mechanism because of their smaller stature [53]. Small stature results in fewer seeds dispersed at longer distances—even for bird-dispersed seeds (Fig 4B). The inability to move seeds far away from the parent tree may force understory plants to be better defended against soil pathogens, which appear to be strong agents of CNDD [1,58,68–70]. Furthermore, negative feedback from soil pathogens may be inversely related to light availability (Smith & Reynolds 2015, Jiang et al. 2020) [71,72]. Many understory plants are naturally well defended because of the importance of preserving plant tissue in a low-light environment [73,74]; thus, understory plants may be predisposed to developing greater defenses to pathogens rather than increasing dispersal abilities.

Differences in the level of overdispersion between canopy species and understory species did not appear to be due to the spatial scale of study in spite of our relatively small plot size. If spatial scale had biased our results, we would have expected the spatial point pattern analysis to show little evidence of overdispersion for large canopy trees, but rather a signature indistinguishable from complete spatial random. Furthermore, Zhu *et al.* [30] demonstrated that when NDD is present it is most likely to be present at the 0–5 m scale and peaks at 5 m (see also [29]). Our results showed a clear spatial signature of overdispersion for our largest individuals. Thus, it seems unlikely that our findings were caused by differences in plant scale. Furthermore, Bagchi & Illian [46] demonstrate that replicated point pattern analysis is significantly more robust to problems of small scale than traditional point pattern analysis.

## Conclusions

The intense focus on canopy trees, and in particular on tree seedlings, may bias the current understanding of diversity maintenance in forest ecosystems [25,26]. If we had restricted our sampling to only the smallest understory individuals, we would have concluded that CNDD maintains woody understory plant diversity but not canopy tree diversity. However, examining larger individuals indicated that adult canopy trees became overdispersed as they matured, but that understory plants did not. Zhu and colleagues [18], Detto and colleagues [26] and Yao and colleagues [25] all emphasized similar caution in drawing large-scale conclusions from studies of seedling dynamics for three reasons. First, patterns of seedling mortality often have little effect on broader community and demographic patterns [18]. Second, NDD tends to decrease with ontogeny rather than increase [25]. Finally, studies of NDD at the recruitment level may overestimate NDD due to regression dilution ([26]; but see [4]).

To fully understand the maintenance of plant species diversity, it is necessary to examine spatial patterns across plant sizes, as well as across plant groups that vary in life history strategies. Spatial patterns may be even more complex when considering species that vary more broadly in their life history strategies, such as herbaceous species, which comprise the majority

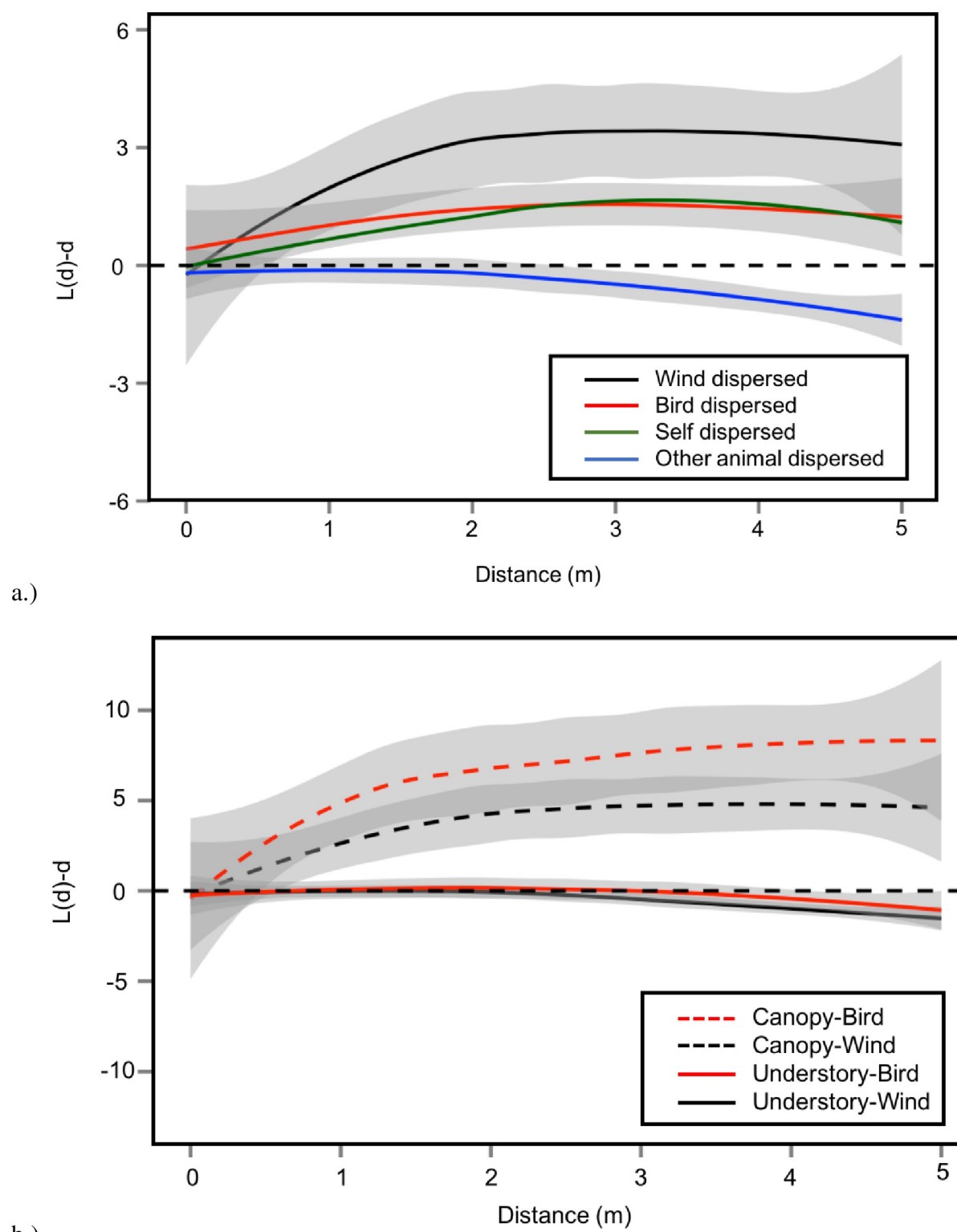

**Fig 4. Pooled Besag's L statistic across distance from spatial point pattern analysis of the woody plant community stratified by dispersal mechanism and plant type at Powdermill Nature Reserve in southwestern Pennsylvania.** a.) Wind dispersed(n = 16), bird dispersed (n = 23), and self dispersed(n = 2) species were significantly more overdispersed than species dispersed by animals other than birds ($n_{animal}$ = 6). b.) Canopy plants were significantly more overdispersed than understory plants regardless of dispersal mechanism($n_{canopy-bird}$ = 5, $n_{canopy-wind}$ = 12, $n_{understory-bird}$ = 18, $n_{understory-wind}$ = 5). Bird dispersal was emphasized here; however, plants did not differ significantly

from wind dispersed plants either in the canopy and the understory. All reported sample sizes (n) are in number of total point patterns contributing to a pooled L function. Grey shaded regions represent 95% confidence intervals, darker grey regions represent overlapping confidence intervals. When confidence intervals overlap, we consider two point patterns to be the same in the overlapping region. We consider point patterns where the confidence intervals overlap with the black dotted line to not be significantly different from complete spatial random in that region.

of plant diversity in temperate deciduous forests [63] and are largely neglected with respect to their diversity maintenance [17]. Nevertheless, even by simply dividing the woody plant community into canopy trees and woody understory plants, we demonstrate that CNDD, which appears to maintain canopy tree diversity, may not be strong enough to overcome dispersal limitation and maintain understory woody plant diversity in this temperate forest.

## Supporting information

**S1 Table. Linear estimates of the relationship between L and distance for all pooled point patterns at Powdermill Nature Reserve.** We report any pooled point pattern as overdispersed if it has a significantly positive slope and any pooled point pattern as clustered if it has a significantly negative slope.
(DOCX)

**S2 Table. Linear estimates of the relationship between L and distance for all pooled point patterns utilizing degrees of freedom based on the number of points represented by each point pattern rather than the degrees of freedom based on the number of L estimates.** To make a more conservative estimate of significance, we calculated the P value for each pooled point pattern using the standard deviation of the L estimates and the number of points contributing to each point pattern. We used the number of points contributing rather than the number of L estimates because L is calculated 51 times (each 10 cm distance bin) for each individual point pattern resulting in an inflated degrees of freedom for the overall model. We then calculated the t statistic as the slope/standard error and used a T table to find the estimated P value for a two-tailed t test. We report the P value for each T statistic at the closest degrees of freedom on the table to our degrees of freedom that was not greater than the actual degrees of freedom (i.e. for a degrees of freedom of 204, we report the p-value for 200 degrees of freedom). This analysis may be overly conservative because the variance, standard deviation, and standard error are calculated based on the L estimates which have a higher variance (as they are calculated 51 times per point) than the average L estimate for each point.
(DOCX)

**S3 Table. List of species from Powdermill Nature Reserve.** We classified species using the Flora of North America species descriptions. If a species had an average height of 5 m or higher, we classified it as a canopy species. If a species had an average height of 5 m or lower, we classified it as an understory species. We based our dispersal syndrome on the description of seed morphology.
(DOCX)

## Acknowledgments

The authors would like to thank Robert Bagchi for helpful comments on the manuscript. The authors would also like to thank Arie Hunt and Joe Strini for field assistance, Jacob Slyder and James Whitacre for GIS and GPS assistance, Cokie Lindsay for administrative and tactical support, and John Wenzel for input on study design and statistical efforts as well as general

support. Thanks also to M. Elizabeth Rodriguez Ronderos, Sergio Estrada Villegas, and Sasha Wright for comments on early drafts of this manuscript.

## Author Contributions

**Conceptualization:** Kathryn E. Barry, Stefan A. Schnitzer.

**Data curation:** Kathryn E. Barry.

**Formal analysis:** Kathryn E. Barry.

**Funding acquisition:** Kathryn E. Barry, Stefan A. Schnitzer.

**Investigation:** Kathryn E. Barry.

**Methodology:** Kathryn E. Barry, Stefan A. Schnitzer.

**Project administration:** Kathryn E. Barry, Stefan A. Schnitzer.

**Supervision:** Stefan A. Schnitzer.

**Validation:** Kathryn E. Barry.

**Visualization:** Kathryn E. Barry.

**Writing – original draft:** Kathryn E. Barry.

**Writing – review & editing:** Kathryn E. Barry, Stefan A. Schnitzer.

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
