## [Decision Letter · Decision Letter 0]

19 Mar 2021

PONE-D-20-40957

Are we missing the forest for the trees? Conspecific negative density dependence in a temperate deciduous forest

PLOS ONE

Dear Dr. Barry,

Thank you for submitting your manuscript to PLOS ONE. After careful consideration, we feel that it has merit but does not fully meet PLOS ONE’s publication criteria as it currently stands. Therefore, we invite you to submit a revised version of the manuscript that addresses the points raised during the review process.

ACADEMIC EDITOR:Please make more clarifications on how to explain the possible reasons of your results

We look forward to receiving your revised manuscript.

Kind regards,

RunGuo Zang

Academic Editor

PLOS ONE

Journal Requirements:

Additional Editor Comments (if provided):

Please make revisions according to the concerns of the reviewers

Reviewers' comments:

Reviewer's Responses to Questions

**Comments to the Author**

1. Is the manuscript technically sound, and do the data support the conclusions?

Reviewer #1: No

Reviewer #2: Yes

2. Has the statistical analysis been performed appropriately and rigorously? 

Reviewer #1: No

Reviewer #2: Yes

3. Have the authors made all data underlying the findings in their manuscript fully available?

Reviewer #1: No

Reviewer #2: Yes

4. Is the manuscript presented in an intelligible fashion and written in standard English?

Reviewer #1: Yes

Reviewer #2: Yes

5. Review Comments to the Author

Reviewer #1: This manuscript studied three conditions that they claim necessary for conspecific negative density dependence (CNDD) to maintain species coexistence by conducting plot surveys in a temperate forest and using spatial point pattern analyses. The study species included different plant growth forms (although results for mid-story trees and lianas were not reported in this manuscript), growth stages, and dispersal modes. Results showed that plants were overdispersed overall, which was a pattern driven by larger growth stages of canopy tree species (“adult canopy trees”) but not understory plants. Because understory plant species can make up the majority of species composition in temperate forest plant diversity, focusing on trees to draw conclusions on CNDD as a mechanism to maintain diversity in forest communities would overestimate its importance where the importance of CNDD in maintaining species diversity.

I liked how this study included growth stages beyond seedlings, which most studies on CNDD are focused on, as I also agree that effects of small seedling mortality may be limited on broader community dynamics. However, there are two major concerns.

First, the framework of this study is not well integrated in the context of existing (a large number of) literature on the topic. It is not a bad idea to test the three conditions (i.e, most individuals will be overdispersed due to CNDD, the degree of CNDD should increase with growth stage due to compounding effects, and CNDD will operate across species with different life history strategies). However, these three conditions are not necessary or sufficient for CNDD to promote species coexistence. This study appears to assume that overdispersion is a result of CNDD (the first condition) and use overdispersion to detect CNDD in the second and third conditions but as one of the key ideas in the manuscript, not sufficient mechanism and rationale connecting the analyses and CNDD are not provided (e.g., see a paper by Gray & He 2009 Forest Ecology and Management). In addition, even without CNDD, one would expect overdispersed patterns in larger individuals because older (and likely larger) individuals are more likely to have lost its true parent trees, which are more likely to be their closest adults (due to initial clumping and dispersal limitation).

In addition, I was wondering why this manuscript never mentioned that (1) most tropical species are clumped and not overdispersed (e.g., Hubbell 1979 Science, which this manuscript cites but I believe miss-cited in line 65; Condit et al. 2000 Science; Armestro et al. 1986 Biotropica, which includes temperate forests); (2) for CNDD to promote species diversity, common species should suffer stronger CNDD than rare species because species diversity is inherently related to species’ rarity (tropical forest diversity is possible by having many rare species and a handful of common species); and (3) there is accumulating evidence (e.g., Bennett et al. 2017 Science; Jiang et al. 2021 Ecology) that the strength and the sign of density dependence can be largely determined by the type of mycorrhizal association plants have. This may also explain less aggregated spatial patterns in temperate forests (e.g., Armestro et al. 1986 Biotropica). Many species included in this study (e.g., Betula, Carpinus, Carya, Fagus, Quercus) are ectomycorrhizal (unlike many tropical species), which often show less CNDD or positive density dependence.

Second concern relates to the sampling method used in this study. Although there is brief rationale about using small plot sizes (Lines 132-141; 289-298), I am really not sure how one can conduct rigorous spatial point pattern analyses using only 16 plots of only 10 m diameter (which is in total 0.13 ha, even when it is across a bigger spatial scale). One cannot even calculate distances beyond 10 m (and much smaller for larger trees) within a plot. Some large trees can have a crown size greater than 10 m but was the forest where the study took place consists of all small trees? Were distances calculated among plots? Using a mean 5 m height as a cutoff of “canopy” tree (lines 566-567) makes me wonder (a shrub can easily be 5+ m tall)... If “adult canopy trees” (line 36) are trees 5-10 m tall, then I do not think “canopy trees” in this manuscript is what most readers would be thinking. In line 135, it cites three papers but none of them used such a small plot sampling scheme and use plots greater than 24 ha. I am guessing this was a side project of another bigger project but for a project trying to cover different growth stages, the study design is flawed. That is why the distance in the figures are up to 6 m, but I am not convinced that any conclusion made at such a small spatial scale can be extrapolated to any dynamics at a larger scale. Perhaps, would that explain some of the inconsistencies in results with other previous studies? Also, at such a small spatial scale, existence of a plot (and out of only 16 plots) that happened to have included a reproductive tree would heavily bias the results (and avoiding them intentionally would also likely bias the included trees to be smaller). Excluding gaps may also bias the results and limit the generality of the findings because species associated with gaps can be rare.

Minor notes

Page 4: This is a field research but details are not provided.

Line 69: What is a “rare species effect”?

Lines 117-146: When was the field survey conducted and over what time frame?

Line 106: How was each growth stage defined? Is it different from the height classes (Lines142-143)? How were these size classes determined?

Line 318: CNDD is known to be strong in grasses; how do you reconcile it you’re your results?

Figure 3: What explains “self-dispersed” not to be the most aggregated compared to other dispersal modes?

Reviewer #2: PLOS ONE

Barry & Schnitzer 2021—Manuscript # PONE-D-20-40957

Overall statement:

This manuscript used spatially explicit plant survey data from a temperate deciduous forest to test conspecific negative density dependence. It is important research which addresses the (lack of!) generality of a leading ecological theory for diversity maintenance. The novelty of the study lies in the system: the vast majority of CNDD studies come from tree data, whereas this study looks at all woody understory species (including vines and shrubs). However, prior to publication in PLOS ONE, I have a few suggestions to clarify the manuscript and put the results into context:

1. Do not over-reach your results in the intro or discussion, because herbaceous plants were not included in the surveys. Your study is a great addition to the CNDD literature because most studies come from trees, but because you did not include herbaceous species (the most species-rich growth form in your study system), I would be careful not to be too broad in calling the analyzed community the whole understory. Your main point/contribution is still true, just modify by saying “woody understory”.

2. Provide justification for your height classes, because they seem somewhat arbitrary to compare among different plant growth-forms (is height a good proxy for ontogeny to compare trees and non-trees?)

3. Overall the discussion could use a little more depth of digging into the mechanisms or the “why” of the results

I also have included comments below for minor changes to improve clarity.

In-line comments and section-specific recommendations:

Abstract:

In-line comments

37 I would be careful here—using that “80% of plant diversity” implies you looked at herbaceous species, which you didn’t. Shrubs make up more like ~10% of the temperate forest vascular plant species, which is still more than trees (~7%) (see Gilliam 2007 BioScience, Spicer et al. 2020 Ecology). I’d temper this argument in the abstract to just make the argument that CNDD has never (? scarcely?) been tested in any growth form other than trees and lianas. You can still easily argue that trees make up a tiny minority of the species in temperate forests.

Introduction:

In-line comments

47 Typo in LaManna’s name in citation

79-89 Rephrase a little unclear here

90 It might be good to guide the reader specifically why growing near a conspecific adult would get worse over time.

95 Be more specific in what you mean by 20% of the community

99-102 Other than the “most species aren’t overstory trees” argument, I think you need to argue why theoretically we expect CNDD to be stronger (or weaker) in non-tree growth forms. You hint at shorter dispersal distances, so would that translate to stronger or weaker NDD? Expand a little more.

Methods

General comments

The only hesitation I have with your methods broadly is your height classes. Assuming you’re using this as a proxy for ontogenic stage, do we know if different growth forms should be grouped in the same height classes? Couldn’t a reproducing adult shrub be <0.5m? And might vine and tree seedlings grow at very different rates? Just wondering if there is a better (but still reasonable) proxy, or if you could divide into ontogenic stages more directly. If not, please justify the height categories (assumedly based on literature from trees), make an argument for why they should be good approximations for shrubs and understory trees too, and make sure to connect the reader to the “why”—will the same height categories be competing?

I think one of your supplementals should be a list of the species with their life-history traits. This would be useful for future studies and to clarify how many species of each category were in the forest.

I also ask for one clarification for the statistical analyses: specify that you put all factors in one model (which is clear by looking at the tables, but less clear in the methods). As written, because there are several “levels” at which the analyses were run: with all plants combined, with individuals separated by size, by growth form, and by dispersal mechanisms, those could each be separate models. One sentence would suffice to explicitly state the variables in the model.

In-line comments

139 Do you have the actual range of accuracy (when taking the GPS points), rather than “up to 10cm”? (Especially if the seedlings were closer together than your less-accurate readings were)

142 In your intro you explicitly said “throughout ontogeny”. Be specific here and say you are using plant size as a proxy for age. Do we know how valid this is for shrubs? Are there good data on how shrub size changes through ontogeny? (See general comments above)

144-145 I like this dispersal syndrome approach to understanding why you see the patterns, but I feel like you missed an opportunity to connect to theory/expectation. Which groups would you predict to have stronger CNDD? How much do these vary within or between growth form groups (so would they be confounded)?

152 I would suggest adding a real quick phrase to justify/explain L (e.g. “for ease of interpretation”)

155 Did the cutoff for removing a species have to be 5 individuals per plot, or total? Were any plots “empty” (no species with >5 individuals/species)? The parenthetical statement makes it sound like there were originally more than 16 plots, but the earlier section says there were 16 originally (I would just take the parenthetical out here if so). If not, how many plots were removed? Or just have a short statement on not analyzing “empty” plots.

157-158 By “species type” do you mean growth form (or life-history strategies)? Stay consistent with terminology or specify your categories somewhere in the methods section. In the introduction (lines 105-106), you specify five types: “shrubs, understory trees, mid-story trees, canopy trees, and lianas”, but just report “canopy trees” versus “understory plants”. How did you categorize them? This is listed in Table S2, but it should more explicitly be referred to in the methods.

161 Couldn’t complete random be a possibility (not likely, but possible)? If so, replace “to ensure” with “to compare to complete random”—isn’t that the null model?

166-188 The authors made a noble attempt to explain these nuanced predictions and justify their interpretation of L(d)-d; it still takes the reader on a bit of a roller coaster. Would it be possible to just put in a supplemental figure that shows a predictions table/figure? This seems so much easier to see rather than imagining from pretty technical prose.

225-227 This is a really good clarification (but maybe belongs in the methods?)

Results

General comments:

Compelling results, interesting, and well-displayed.

Discussion

General comments:

Overall, I wanted the discussion to dig into the mechanisms more, and further explore why we might expect CNDD to be less important for understory plants than overstory trees. What did we learn from the height classes versus the growth form analysis? Might also be interesting to mention the non-native species and just see if they are doing anything different, or discuss expectations for more growth forms (shrubs vs. understory trees versus expectations for lianas or herbs).

Also, why does the shape of the pattern (L(d)-d vs. Distance; Figure 2b) look so different for >5m trees versus the others? Discuss the biological significance/interpretation of the <0.5m trees having zero L(d)-d?

In-line comments

259-273 These two paragraphs seem like too many sentences to say “mechanisms other than CNDD are at play” without actually suggestion what they may be. Expand by suggesting what mechanisms are most likely driving the understory patterns, or eliminate because the next paragraphs get into the mechanisms for trees.

Conclusions

Conclusion provides a nice synthesis of the main results and implications.

In-line comments

305-306 This specific results sentence is not necessary for the conclusion.

Figures

Table 1. Can you put the biological interpretation of a significant L statistic right here in the caption (like in Supplemental Table 1)? It also might help to bold the significant ones.

Figure 2a,b. Even though the colors correspond to 2a, change the labels somewhere on the actual figure to say “overstory” (2b) and “understory” (2c). It also might help interpretation to remind the reader in the figure caption what the biological interpretation is of the 95% CI shaded regions overlapping each other versus overlapping 0 (rather than just say what the conclusion is).

Figure 3b. I would pick another line style for the zero line (or for the canopy-wind) because they are the same.

Table S1: This table would be easier to quick glean the message if the overdispersed or clustered values were emphasized differently (e.g., overdispersed bolded, clustered italicized). As is, the reader has to look at both the t-stat and the p value to interpret.

Table S2: This was only ever referenced once, and not explained at all. Maybe one quick sentence in the methods “we confirmed…with an even more conservative estimate”

Table S2: Were you really able to tell the Carya and Fraxinus species apart at such small life stages? Typo: de-capitalize Americana for Fraxinus, and isn’t the common name white ash?

566 An average height of 5m or higher from your dataset, or reported in the Flora of North America? Would be good to mention because one could also argue for max height, rather than average, being a good metric of over- versus understory status (can it ever break through the understory, versus does the species on average?).

6. PLOS authors have the option to publish the peer review history of their article (what does this mean?). If published, this will include your full peer review and any attached files.

Reviewer #1: No

Reviewer #2: No

---

## [Author Response · Author response to Decision Letter 0]

14 Jun 2021

We have also submitted this information as a .docx with the rest of our materials where the formatting makes it a bit easier to see our response to individual comments. 

PONE-D-20-40957

Are we missing the forest for the trees? Conspecific negative density dependence in a temperate deciduous forest

PLOS ONE

Dear Dr. Barry,

Thank you for submitting your manuscript to PLOS ONE. After careful consideration, we feel that it has merit but does not fully meet PLOS ONE’s publication criteria as it currently stands. Therefore, we invite you to submit a revised version of the manuscript that addresses the points raised during the review process.

ACADEMIC EDITOR:Please make more clarifications on how to explain the possible reasons of your results

We look forward to receiving your revised manuscript.

Kind regards,

RunGuo Zang

Academic Editor

PLOS ONE

Journal Requirements:

Additional Editor Comments (if provided):

Please make revisions according to the concerns of the reviewers

Many thanks again to the editor for their constructive oversight over this process. We have numbered the comments of the reviewers for ease of response and respond to them directly below the comment.

Reviewer #1: This manuscript studied three conditions that they claim necessary for conspecific negative density dependence (CNDD) to maintain species coexistence by conducting plot surveys in a temperate forest and using spatial point pattern analyses. The study species included different plant growth forms (although results for mid-story trees and lianas were not reported in this manuscript), growth stages, and dispersal modes. Results showed that plants were overdispersed overall, which was a pattern driven by larger growth stages of canopy tree species (“adult canopy trees”) but not understory plants. Because understory plant species can make up the majority of species composition in temperate forest plant diversity, focusing on trees to draw conclusions on CNDD as a mechanism to maintain diversity in forest communities would overestimate its importance where the importance of CNDD in maintaining species diversity.

I liked how this study included growth stages beyond seedlings, which most studies on CNDD are focused on, as I also agree that effects of small seedling mortality may be limited on broader community dynamics. However, there are two major concerns.

Many thanks to the review for their excellent summary of this work and for their kind words regarding the inclusion of different growth stages. We have now numbered their comments for ease of response and respond to them in bold. 

1. First, the framework of this study is not well integrated in the context of existing (a large number of) literature on the topic. It is not a bad idea to test the three conditions (i.e, most individuals will be overdispersed due to CNDD, the degree of CNDD should increase with growth stage due to compounding effects, and CNDD will operate across species with different life history strategies). However, these three conditions are not necessary or sufficient for CNDD to promote species coexistence. This study appears to assume that overdispersion is a result of CNDD (the first condition) and use overdispersion to detect CNDD in the second and third conditions but as one of the key ideas in the manuscript, not sufficient mechanism and rationale connecting the analyses and CNDD are not provided (e.g., see a paper by Gray & He 2009 Forest Ecology and Management). In addition, even without CNDD, one would expect overdispersed patterns in larger individuals because older (and likely larger) individuals are more likely to have lost its true parent trees, which are more likely to be their closest adults (due to initial clumping and dispersal limitation).

First, we thank the reviewer for agreeing that it is a good idea to test the three conditions responsible for CNDD. The reviewer raises a good point about other potential drivers of overdispersion of larger stems and we have added a paragraph to the Discussion stating that the increase in the probability of over dispersion with plant size could have several potential explanations (Lines 237 to 239) which read: 

“However, we cannot rule out the possibility that trees may be more likely to be overdispersed with size simply because the larger (and presumably older) the tree the greater the probability of mortality for the parent (which is often nearby) [59].”

Nonetheless, we appreciate that the reviewer acknowledges the benefits of examining multiple plant size classes. In addition, we now state that these three conditions are consistent with CNDD; however, our approach goes further than other studies that typically test the over dispersion component of CNDD. 

2. In addition, I was wondering why this manuscript never mentioned that (1) most tropical species are clumped and not overdispersed (e.g., Hubbell 1979 Science, which this manuscript cites but I believe miss-cited in line 65; Condit et al. 2000 Science; Armestro et al. 1986 Biotropica, which includes temperate forests); (2) for CNDD to promote species diversity, common species should suffer stronger CNDD than rare species because species diversity is inherently related to species’ rarity (tropical forest diversity is possible by having many rare species and a handful of common species); and (3) there is accumulating evidence (e.g., Bennett et al. 2017 Science; Jiang et al. 2021 Ecology) that the strength and the sign of density dependence can be largely determined by the type of mycorrhizal association plants have. This may also explain less aggregated spatial patterns in temperate forests (e.g., Armestro et al. 1986 Biotropica). Many species included in this study (e.g., Betula, Carpinus, Carya, Fagus, Quercus) are ectomycorrhizal (unlike many tropical species), which often show less CNDD or positive density dependence.

We also address the reviewer’s comment that tree species distribution is mostly clumped. The spatial pattern of tree distribution is likely related to scale. Several studies that show that tropical tree seedling and sapling performance and distribution are over dispersed in relation to conspecifics (e.g., Comita et al. 2010, Ledo & Schnitzer 2014), but that at larger scales these same species may appear clumped. We now clarify this on lines 81 to 85) which read: 

“Currently, the evidence for CNDD beyond the seed to seedling transition is mixed. For example, a study by Yao et al. [25] found that CNDD decreased with increasing tree ontogeny in a temperate forest. In fact, many species in both temperate and tropical forests do not have an overdispersed distribution [19,28]. By contrast, Guo et al. [29] found that 75% of tree species demonstrated CNDD as adults in subtropical forests (see also [30–32]).”

Another point raised by the reviewer is that common species suffer more from CNDD than rare species. However, several influential studies have shown the opposite: that adult species frequency is negatively correlated with the strength of CNDD (e.g., Comita et al. 2010, Mangan et al. 2010, Johnson et al. 2012). That is, rare species suffer more from proximity to conspecific adults than common species. We agree with the reviewer that, theoretically, common species should have strongest CNDD; nonetheless, empirical data do not support this expectation.

3. Second concern relates to the sampling method used in this study. Although there is brief rationale about using small plot sizes (Lines 132-141; 289-298), I am really not sure how one can conduct rigorous spatial point pattern analyses using only 16 plots of only 10 m diameter (which is in total 0.13 ha, even when it is across a bigger spatial scale). One cannot even calculate distances beyond 10 m (and much smaller for larger trees) within a plot. Some large trees can have a crown size greater than 10 m but was the forest where the study took place consists of all small trees? Were distances calculated among plots? Using a mean 5 m height as a cutoff of “canopy” tree (lines 566-567) makes me wonder (a shrub can easily be 5+ m tall)... If “adult canopy trees” (line 36) are trees 5-10 m tall, then I do not think “canopy trees” in this manuscript is what most readers would be thinking. In line 135, it cites three papers but none of them used such a small plot sampling scheme and use plots greater than 24 ha. I am guessing this was a side project of another bigger project but for a project trying to cover different growth stages, the study design is flawed. That is why the distance in the figures are up to 6 m, but I am not convinced that any conclusion made at such a small spatial scale can be extrapolated to any dynamics at a larger scale. Perhaps, would that explain some of the inconsistencies in results with other previous studies? Also, at such a small spatial scale, existence of a plot (and out of only 16 plots) that happened to have included a reproductive tree would heavily bias the results (and avoiding them intentionally would also likely bias the included trees to be smaller). Excluding gaps may also bias the results and limit the generality of the findings because species associated with gaps can be rare.

Second, we appreciate the reviewer’s concerns about the size and extent of our study. We believe that our findings are robust based on our sampling scheme. However, we now directly address the sampling issues raised by the reviewer and we discuss how the size of the study could have influenced our results (300-309) which read: 

“Differences in the level of overdispersion between canopy species and understory species did not appear to be due to the spatial scale of study in spite of our relatively small plot size. If spatial scale had biased our results, we would have expected the spatial point pattern analysis to show little evidence of overdispersion for large canopy trees, but rather a signature indistinguishable from complete spatial random. Furthermore, Zhu et al. [30] demonstrated that when NDD is present it is most likely to be present at the 0-5 m scale and peaks at 5 m (see also [29]). Our results showed a clear spatial signature of overdispersion for our largest individuals. Thus, it seems unlikely that our findings were caused by differences in plant scale. Furthermore, Bagchi & Illian [46] demonstrate that replicated point pattern analysis is significantly more robust to problems of small scale than traditional point pattern analysis.”

Overall, our study was large enough for the results to be consistent with CNDD and we believe that a larger sample size would likely not change our results or interpretations. However, in addition to addressing these concerns in the manuscript, we have also tempered the interpretation of our findings. 

Minor notes

Page 4: This is a field research but details are not provided.

We could not be quite sure what the reviewer was referring to. We have now clarified on line 104 that this research was done in the field. 

Line 69: What is a “rare species effect”?

We have rephrased this to now say that “CNDD will not theoretically benefit rare species…” (on line 59 of the revised manuscript) which we hope clarifies what we mean by a rare species effect. 

Lines 117-146: When was the field survey conducted and over what time frame?

We have now clarified this on line 127, we established these plots in May and June of 2014. 

Line 106: How was each growth stage defined? Is it different from the height classes (Lines142-143)? How were these size classes determined?

The use of height classes is now clarified on lines 138 to 142. 

Line 318: CNDD is known to be strong in grasses; how do you reconcile it you’re your results?

We have clarified in this version of our paper that our results refer only to woody plants in temperate deciduous forests. They would therefore not be in conflict with results from grasslands. 

Figure 3: What explains “self-dispersed” not to be the most aggregated compared to other dispersal modes?

This is a great question! The most likely explanation is that our self-dispersed species managed to get their seeds farther away than plants that were dispersed by some types of animals. Hamamelis virginiana may not be typical of other self-dispersed species because it uses a ballistic method of dispersal so seeds may travel further than expected for self-dispersed species. 

 

Reviewer #2: 

This manuscript used spatially explicit plant survey data from a temperate deciduous forest to test conspecific negative density dependence. It is important research which addresses the (lack of!) generality of a leading ecological theory for diversity maintenance. The novelty of the study lies in the system: the vast majority of CNDD studies come from tree data, whereas this study looks at all woody understory species (including vines and shrubs). 

First, we would like to extend our many thanks to the reviewer for their overall positive thoughts on the manuscript. We found their comments to be positive, constructive, and detailed and feel that they have contributed to a much stronger manuscript upon resubmission. 

However, prior to publication in PLOS ONE, I have a few suggestions to clarify the manuscript and put the results into context:

1. Do not over-reach your results in the intro or discussion, because herbaceous plants were not included in the surveys. Your study is a great addition to the CNDD literature because most studies come from trees, but because you did not include herbaceous species (the most species-rich growth form in your study system), I would be careful not to be too broad in calling the analyzed community the whole understory. Your main point/contribution is still true, just modify by saying “woody understory”.

We have now tried to clarify our contribution throughout the abstract, introduction, and discussion. These revisions are small but impactful. For example, throughout we’ve emphasized that we are speaking about the woody understory rather than the “understory” as we previously wrote. 

2. Provide justification for your height classes, because they seem somewhat arbitrary to compare among different plant growth-forms (is height a good proxy for ontogeny to compare trees and non-trees?)

Thanks to the reviewer for pointing out that this was unclear. We use height classes as a proxy for both relative age (assuming that woody plants get taller as they get older) and also their position in the forest (i.e. are they a in the canopy or in the understory). We use height in this way because it provides a different type of information from just understory vs. overstory or different dispersal syndromes. A small individual of an overstory species likely has a smaller seed shadow than a taller individual of the same species. Because we believe dispersal to be so closely intertwined with CNDD, we felt that height was likely an important factor. We have now clarified this in two places in the manuscript. 

A.) In the introduction on lines 89 to 94. 

B.) More in depth in the methods on lines 138 to 142 where it now states: “We use these height classes as a proxy for both relative age (assuming that plants get taller as they get older) and position within the forest. Individuals that are shorter are less likely to be able to disperse seeds farther away than individuals that are taller even if they belong to the same species and have the same dispersal syndrome.”

3. Overall the discussion could use a little more depth of digging into the mechanisms or the “why” of the results

Thanks to the reviewer for this comment as we believe it significantly improves the discussion of this paper. We have revised the discussion to better discuss the “why” of our results. This is especially evident in the two paragraphs highlighted by the reviewer in their specific comments on the discussion. These two paragraphs have now been significantly revised in two ways: 

1. We’ve removed some of the redundant language from these paragraphs to make the discussion of alternative mechanisms a bit more streamlined. 

2. We’ve added more specific discussion of what other mechanisms may be more relevant. 

These paragraphs now read (on lines 257 to 284): 

“In temperate forests, CNDD likely does not occur in isolation. Rather, CNDD and other mechanisms like facilitation, niche specialization, and dispersal limitation likely interact to maintain diversity in these forests. CNDD may be the most important mechanism for the maintenance of tree species diversity even though these other mechanisms are likely to be occurring simultaneously. But for other plant groups, these other mechanisms like facilitation, niche specialization, and dispersal limitation may be more important relative to CNDD. For example, Ledo and Schnitzer [5], found that clumped spatial distributions may be due to niche specialization in lianas, while trees demonstrated overdispersion indicating that CNDD may be more powerful. Similarly, the relative importance of these different mechanisms may change as plants grow. For example, Yao et al. [25] found that CNDD was important for individuals when they were young and small but that topographic and edaphic factors increased in importance with increasing plant age. Similarly, for tree seedlings invading into a grassland, Wright et al. [64] found that smaller tree seedlings benefited from facilitation in high diversity contexts while larger tree seedlings experienced strong competition. 

At Powdermill Nature Reserve, a similar scenario where overall diversity is maintained by several mechanisms which simultaneously support diversity but also tradeoff in importance depending on the age/size of individuals and their abiotic context. Trees (and especially the largest trees) may be maintained largely by CNDD; whereas, understory plants may be influenced by a number of different mechanisms. There is evidence that CNDD is a weak mechanism for the maintenance of understory plant diversity, since overdispersion is present when understory plants are small (Fig. 2c). However, the lack of overdispersion in larger understory plants indicates that a mechanism (or mechanisms) other than CNDD is a stronger driver of understory plant diversity. Short distance dispersal is often adaptive because site conditions are likely to be the same in the area immediately surrounding a parent plant [65]. Because dispersal syndromes that favor shorter distance dispersal are more common in the understory, mechanisms like niche differentiation that rely on adaptation to specific abiotic factors as found by both Ledo and Schnitzer [5] and Yao et al. [25] may be more important for these understory species.”

Abstract 

1. Line 37 I would be careful here—using that “80% of plant diversity” implies you looked at herbaceous species, which you didn’t. Shrubs make up more like ~10% of the temperate forest vascular plant species, which is still more than trees (~7%) (see Gilliam 2007 BioScience, Spicer et al. 2020 Ecology). I’d temper this argument in the abstract to just make the argument that CNDD has never (? scarcely?) been tested in any growth form other than trees and lianas. You can still easily argue that trees make up a tiny minority of the species in temperate forests.

We’ve now removed this statement and temper this argument throughout. A good example of this can be found on line 32.

Introduction

1. 47 Typo in LaManna’s name in citation

This is now fixed.

2. 79-89 Rephrase a little unclear here

We have now rephrased this paragraph and removed this statement. 

We also revised this partially to accommodate reviewer #2’s major concern #1 as well as reviewer #1’s concern #1. 

3. 90 It might be good to guide the reader specifically why growing near a conspecific adult would get worse over time. 

We have now added this information – this information is now included on lines 86-88. 

4. 95 Be more specific in what you mean by 20% of the community

We have changed this statement also in response to reviewer #2’s comment #1 to say 7% since we are referring to only canopy trees (line 89). 

5. 99-102 Other than the “most species aren’t overstory trees” argument, I think you need to argue why theoretically we expect CNDD to be stronger (or weaker) in non-tree growth forms. You hint at shorter dispersal distances, so would that translate to stronger or weaker NDD? Expand a little more. 

This is now clarified in this paragraph on lines 92 to 98 where it reads: 

“By contrast, understory plants (including understory woody species) represent a larger share of diversity but have a lower capacity for long distance dispersal due to their relatively short stature and position in forest understory. Furthermore, few understory plant species have dispersal syndromes that favor long distance dispersal [33]. Many understory species are gravity dispersed while the majority of temperate canopy trees are wind dispersed. Thus, the strength of CNDD may interact with plant dispersal syndrome.” 

Methods

General comments

The only hesitation I have with your methods broadly is your height classes. Assuming you’re using this as a proxy for ontogenic stage, do we know if different growth forms should be grouped in the same height classes? Couldn’t a reproducing adult shrub be <0.5m? And might vine and tree seedlings grow at very different rates? Just wondering if there is a better (but still reasonable) proxy, or if you could divide into ontogenic stages more directly. If not, please justify the height categories (assumedly based on literature from trees), make an argument for why they should be good approximations for shrubs and understory trees too, and make sure to connect the reader to the “why”—will the same height categories be competing?

We respond to this issue in full in our response to reviewer #2 comment #2. Thanks so much again to the reviewer for highlighting this issue. 

1. I think one of your supplementals should be a list of the species with their life-history traits. This would be useful for future studies and to clarify how many species of each category were in the forest.

This information can be found in table S3. 

2. I also ask for one clarification for the statistical analyses: specify that you put all factors in one model (which is clear by looking at the tables, but less clear in the methods). As written, because there are several “levels” at which the analyses were run: with all plants combined, with individuals separated by size, by growth form, and by dispersal mechanisms, those could each be separate models. One sentence would suffice to explicitly state the variables in the model.

We have now briefly clarified this on line 190 of the methods. 

3. 139 Do you have the actual range of accuracy (when taking the GPS points), rather than “up to 10cm”? (Especially if the seedlings were closer together than your less-accurate readings were)

Unfortunately, we do not have this information. 

4. 142 In your intro you explicitly said “throughout ontogeny”. Be specific here and say you are using plant size as a proxy for age. Do we know how valid this is for shrubs? Are there good data on how shrub size changes through ontogeny? (See general comments above)

We have now clarified this (as detailed above in our response to reviewer #2 comment #1) on lines 138 to 142.

5. 144-145 I like this dispersal syndrome approach to understanding why you see the patterns, but I feel like you missed an opportunity to connect to theory/expectation. Which groups would you predict to have stronger CNDD? How much do these vary within or between growth form groups (so would they be confounded)?

We have tried to clarify this throughout the manuscript. However, we find this a bit of a tough line to walk since our hypothesis seeks to test the generality of CNDD and our original expectation was that CNDD would be common even in understory species. Thus, while we do provide some brief expectations in the introduction in this version of the manuscript as suggested here by Reviewer #2 (see lines 92 to 98), we have largely expounded upon this in the discussion (also to address Reviewer #2’s major comment #3. 

6. 152 I would suggest adding a real quick phrase to justify/explain L (e.g. “for ease of interpretation”)

This phrase is now added on line 150.

7. 155 Did the cutoff for removing a species have to be 5 individuals per plot, or total? Were any plots “empty” (no species with >5 individuals/species)? The parenthetical statement makes it sound like there were originally more than 16 plots, but the earlier section says there were 16 originally (I would just take the parenthetical out here if so). If not, how many plots were removed? Or just have a short statement on not analyzing “empty” plots.

This sentence is now clarified on lines 153-155.

8. 157-158 By “species type” do you mean growth form (or life-history strategies)? Stay consistent with terminology or specify your categories somewhere in the methods section. In the introduction (lines 105-106), you specify five types: “shrubs, understory trees, mid-story trees, canopy trees, and lianas”, but just report “canopy trees” versus “understory plants”. How did you categorize them? This is listed in Table S2, but it should more explicitly be referred to in the methods.

This sentence has now been clarified and we now use growth form throughout to refer only to canopy vs. understory woody plants. 

9. 161 Couldn’t complete random be a possibility (not likely, but possible)? If so, replace “to ensure” with “to compare to complete random”—isn’t that the null model?

This is totally correct. We have now changed this in the manuscript on line 162-163. 

10. 166-188 The authors made a noble attempt to explain these nuanced predictions and justify their interpretation of L(d)-d; it still takes the reader on a bit of a roller coaster. Would it be possible to just put in a supplemental figure that shows a predictions table/figure? This seems so much easier to see rather than imagining from pretty technical prose.

This is a great suggestion and we include now a figure (Figure 1 a,b,c) that does exactly this in the new version of the manuscript. This figure shows a basic conceptual distribution as well as a hypothetical overdispersed and underdispersed distribution. 

11. 225-227 This is a really good clarification (but maybe belongs in the methods?)

Thanks for this suggestion, we’ve now moved it to the methods on lines 157-160. 

Results

General comments:

Compelling results, interesting, and well-displayed.

Thanks so much to the reviewer for including these positive comments as well as the more critical ones! 

Discussion

General comments:

1. Overall, I wanted the discussion to dig into the mechanisms more, and further explore why we might expect CNDD to be less important for understory plants than overstory trees. What did we learn from the height classes versus the growth form analysis? Might also be interesting to mention the non-native species and just see if they are doing anything different, or discuss expectations for more growth forms (shrubs vs. understory trees versus expectations for lianas or herbs).

2. Also, why does the shape of the pattern (L(d)-d vs. Distance; Figure 2b) look so different for >5m trees versus the others? Discuss the biological significance/interpretation of the <0.5m trees having zero L(d)-d?

In-line comments

3. 259-273 These two paragraphs seem like too many sentences to say “mechanisms other than CNDD are at play” without actually suggestion what they may be. Expand by suggesting what mechanisms are most likely driving the understory patterns, or eliminate because the next paragraphs get into the mechanisms for trees.

Thanks to the reviewer for pointing this out. We have now revised this paragraph (see our response to reviewer #2 major comment #3. 

Conclusions

Conclusion provides a nice synthesis of the main results and implications.

In-line comments

305-306 This specific results sentence is not necessary for the conclusion.

We’ve now removed this statement. 

Figures

Table 1. Can you put the biological interpretation of a significant L statistic right here in the caption (like in Supplemental Table 1)? It also might help to bold the significant ones.

Figure 2a,b. Even though the colors correspond to 2a, change the labels somewhere on the actual figure to say “overstory” (2b) and “understory” (2c). It also might help interpretation to remind the reader in the figure caption what the biological interpretation is of the 95% CI shaded regions overlapping each other versus overlapping 0 (rather than just say what the conclusion is).

Thanks to the reviewer for pointing this out, we have accordingly modified the figure and table captions. 

Figure 3b. I would pick another line style for the zero line (or for the canopy-wind) because they are the same.

We’ve tried this a few ways and found it difficult to interpret with a modified line style (since the line styles get increasingly complicated). We’ve tried to clarify this in the figure caption instead and hope this helps. 

Table S1: This table would be easier to quick glean the message if the overdispersed or clustered values were emphasized differently (e.g., overdispersed bolded, clustered italicized). As is, the reader has to look at both the t-stat and the p value to interpret.

While we include these results in the supplement, our interpretation of over- vs. underdispersal depends on the interaction between the results in this table and the figures themselves. For example, if a pattern was significantly overdispersed according to the statistical result but overlapped completely with complete spatial random, we did not consider this a significant result. Our concern with bolding and clustering these results is that it may oversimplify this interpretation and cause more confusion. 

Table S2: This was only ever referenced once, and not explained at all. Maybe one quick sentence in the methods “we confirmed…with an even more conservative estimate”

This has now been added on lines 192-194. 

Table S2: Were you really able to tell the Carya and Fraxinus species apart at such small life stages? Typo: de-capitalize Americana for Fraxinus, and isn’t the common name white ash?

All individuals were >10cm in height and we found that this largely excluded very small seedlings of both Carya and Fraxinus but that we were able to determine these at the species level with a fair degree of accuracy at this height. We consulted with a local botanist when we were unsure of plant identities. We’ve also fixed the typos in this table. 

566 An average height of 5m or higher from your dataset, or reported in the Flora of North America? Would be good to mention because one could also argue for max height, rather than average, being a good metric of over- versus understory status (can it ever break through the understory, versus does the species on average?)

This has now been clarified in the figure caption. We used the average height of the species as reported in the Flora of North America. We agree that max. height may be a potentially better metric, however – this would not change the species that we identify as understory vs. overstory as none of the understory species break through to the canopy in eastern deciduous forests.

---

## [Editor Report · Decision Letter 1]

18 Jun 2021

Are we missing the forest for the trees? Conspecific negative density dependence in a temperate deciduous forest

PONE-D-20-40957R1

Dear Dr. Barry,

We’re pleased to inform you that your manuscript has been judged scientifically suitable for publication and will be formally accepted for publication once it meets all outstanding technical requirements.

Kind regards,

RunGuo Zang

Academic Editor

PLOS ONE

Additional Editor Comments (optional):

Accept
---

## [Editor Report · Acceptance letter]

1 Jul 2021

PONE-D-20-40957R1 

Are we missing the forest for the trees? Conspecific negative density dependence in a temperate deciduous forest 

Dear Dr. Barry:

I'm pleased to inform you that your manuscript has been deemed suitable for publication in PLOS ONE. Congratulations! Your manuscript is now with our production department. 

Kind regards, 

on behalf of

Professor RunGuo Zang 

Academic Editor

PLOS ONE